# A Taxonomic Revision of the Assassin Bug Genus, *Tympanistocoris* (Hemiptera: Reduviidae: Reduviinae)

**DOI:** 10.3390/insects14020165

**Published:** 2023-02-08

**Authors:** Mingyuan Fan, Yingqi Liu, Wanzhi Cai

**Affiliations:** Department of Entomology and MOA Key Lab of Pest Monitoring and Green Management, College of Plant Protection, China Agricultural University, Beijing 100193, China

**Keywords:** *Tympanistocoris*, new species, Papua New Guinea, taxonomy, anastomosing veins

## Abstract

**Simple Summary:**

The assassin bug genus, *Tympanistocoris* Miller, is revised with the redescription of the type species, *T. humilis* Miller, and the description of a new species, *Tympanistocoris usingeri* sp. nov., from Papua New Guinea. The morphological comparation and a key are provided to distinguish the two species. The anastomosing veins on the hemelytra and the systematic position of *Tympanistocoris* are also briefly discussed.

**Abstract:**

The small reduviine genus, *Tympanistocoris* Miller, is revised. The type species of the genus, *T. humilis* Miller, is redescribed, and a new species, *Tympanistocoris usingeri* sp. nov., from Papua New Guinea is described. The illustrations of the antennae, head and pronotum, legs, hemelytra, abdomen, and male genitalia, as well as the habitus of the type specimens are also provided. The new species can be separated from the type species of the genus, *T. humilis* Miller, in the lateral sides of the pronotum with a distinct carina and the posterior margin of the seventh abdominal segment emarginated. The type specimen of the new species is kept in The Natural History Museum, London. The anastomosing veins of the hemelytra and the systematic position of the genus are briefly discussed.

## 1. Introduction

Including at least 140 genera and about 1000 species, Reduviinae is one of the most diverse subfamilies within the family, Reduviidae [1]. Reduviine species are recognized by usually having ocelli, the three-segmented tarsi, two closed cells on the membrane of the hemelytron, and the presence fossula spongiosa on the fore and mid tibiae. In addition, Reduviinae are generally recognized by the absence of specialized morphological features that are characteristic for other subfamilies [2]. Reduviinae has a cosmopolitan distribution, but is more species-rich in the Old and New World tropics. There are also some island endemic genera, such as *Cargasdama* Villiers and *Durevius* Villiers in Madagascar, *Jacobsonocoris* Miller in Sumatra, and *Pheletocoris* Miller in Solomon Islands [1,2]. Moreover, there are many insular genera of Reduviinae which are monotypic, not only reflecting the morphological diversity of this group, but also indicating that the true species diversity needs to be revealed through more detailed studies. New Guinea is a large tropical island with high biodiversity, but the taxonomic revisions of certain reduviine taxa distributed in New Guinea are still lacking.

The assassin bug genus, *Tympanistocoris* (Hemiptera: Reduviidae), was erected by Miller in 1958 to accommodate the reduviine species, T. humilis, from Papua New Guinea [3]. This genus is easily recognized by the large erect processes on the dorsum of the head and pronotum (Figures 1, 2A,B, 5, 6A,B, 7A, 10, 11B, 1 and 13B), the longitudinal carinae on the pronotum (Figures 1, 2B, 5, 6B, 10 and 12), and the special venations of the hemelytra (Figures 1, 2D,E, 5, 7B,C, 10 and 12). Prior to this study, the genus was monotypic [1]. Recent examination of the assassin bug specimens in The Natural History Museum, London, the United Kingdom, revealed a second species of this genus. In the present paper, we describe a new species, *T. usingeri* sp. nov., redescribe the type species of the genus, *Tympanistocoris*, and provide a key to separate them. The illustrations of different body parts, e.g., the antennae, head and pronotum, legs, hemelytra, abdomen, and male genitalia, as well as the habitus of the type specimens, are also provided. The taxonomic value and putative function of the anastomosing veins on the hemelytra and the systematic position of the genus are briefly discussed.

## 2. Materials and Methods

This study is based on the assassin bug specimens kept in The Natural History Museum, London, the United Kingdom (BMNH). Male genitalia were examined by soaking in hot 10% KOH (Sinopharm Chemical Reagent Co., Ltd., Beijing, China) solution for about ten minutes to remove the soft tissue, rinsed in distilled water, and then dissected using a pair of fine tweezers under a binocular microscope (OLYMPUS ZSX7, Japan). The dissected pygophore is stored in a micro vial with glycerol (Sinopharm Chemical Reagent Co., Ltd., Beijing, China) and pinned under the corresponding specimen. All drawings were traced with the aid of a camera lucida (Motic K400, Xiamen, China) using pencils and needle pens. Photos of the habitus of specimens and their labels were taken using a Canon 7D Mark II digital camera (Tokyo, Japan) with a Canon EF 100 mm micro lens. Helicon Focus version 5.3 (Helicon Soft Ltd., Kharkiv, Ukraine) was used for image stacking. Photoshop version CC (Adobe Systems Inc, USA) was used for the image processing and layout of figure plates. Morphological terminology mainly follows those of Lent and Wygodzinsky [4] and Weirauch [5]. Measurements were obtained using a calibrated micrometer. The specimen was measured from dorsal view in the resting condition, and we also adjusted the angles to make sure that the antennae were also measured from the dorsal view. All measurements are in millimeters. The body length was measured from the tip of head to the apex of the hemelytron in resting condition and the maximum width of the abdomen was measured with the posterior connexival segment processes excluded. Information from each label is placed between quotation marks; the shape, color, and writing of the labels are indicated in square brackets and handwritten labels are denoted as “hw”, whereas printed ones are denoted as “pr”.

## 3. Results

### Taxonomy

Order Hemiptera Linnaeus, 1758.Suborder Heteroptera Latreille, 1810.Infraorder Cimicomorpha Leston, Pendergrast & Southwood, 1954.Family Reduviidae Latreille, 1807.Subfamily Reduviinae Amyot & Serville, 1843.**Genus *Tympanistocoris* Miller, 1958** (Figure 1, Figure 2, Figure 3, Figure 4, Figure 5, Figure 6, Figure 7, Figure 8, Figure 9, Figure 10, Figure 11, Figure 12 and Figure 13)

*Tympanistocoris* Miller, 1958: 74; Maldonado-Capriles, 1990: 455.

**Type species:***Tympanistocoris humilis* Miller, 1958, by original designation.

**Diagnosis:** This genus closely resembles to the Australian reduviine genus, *Noualhierana* Miller, in the head and thoracic structures. However, it can be easily separated from the latter by the large erect processes on the dorsum of the head and pronotum (Figure 1, Figure 2A,B, Figure 5, Figure 6A,B, Figure 7A, Figure 10, Figure 11B, Figure 12 and Figure 13B), the longitudinal carinae on the pronotum, the humeral angles tuberculately produced (Figure 1, Figure 2B, Figure 5, Figure 6A, Figure 10 and Figure 12), and the posterior angles of the connexival segments protruded (Figure 1, Figure 3A, Figure 5, Figure 8B, Figure 10, Figure 11A, Figure 12 and Figure 13A).

**Figure 1 insects-14-00165-f001:**
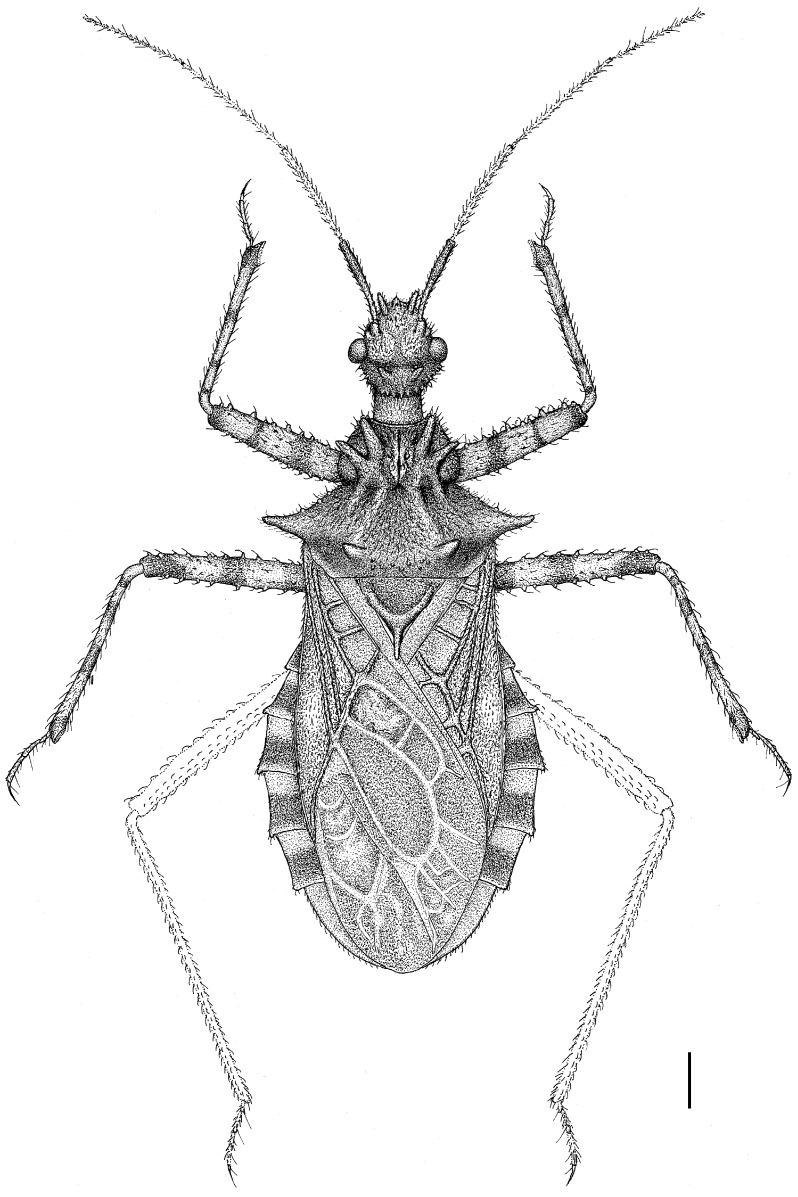
*Tympanistocoris humilis* Miller, male. Holotype, habitus. Scale bar = 1 mm. The apical part of scutellum was destroyed, and it was drawn based on that of *T. usingeri* sp. nov.

**Redescription:** Small size (body length, 11.3–13.6 mm), body elliptical, robust, with many setigerous tubercles on head, pronotum, and legs. Male macropterous form, female unknown. Head slightly longer than wide, much shorter than pronotum, transverse sulcus situated behind eyes, anteocular portion of head longer than postocular portion, anteocular and postocular portions each with a pair of erect tuberculate processes, one pair located between antennophores and one pair located immediately behind ocelli; antennophores closer to eyes than to apex of head; first antennal segment longer than anteocular portion, third antennal segment longest; eye spherical, interocular distance over two times of eye diameter in dorsal view, reniform and not reaching upper and lower margins of head in lateral view; ocelli widely separated, interocellar distance over three times of width of ocellus; labium thick, first visible labial segment thickest, subequal to second segment in length, third segment distinctly shorter than first and second segments. Pronotum wider than long; anterior pronotal lobe much narrower than posterior pronotal lobe, with two pairs of erect long tuberculate processes; posterior pronotal lobe with two long and two short carinae, and one pair of long tuberculate processes before posterior margin; humeral angles tuberculately produced. Scutellum nearly equilateral-triangle-shaped, apex of scutellar process produced. Meso- and metasternum, and second abdominal sternum with a median, longitudinal carina ventrally; mesosternum with a median, short tuberculate process. Hemelytron with two cells on membrane, inner cell much smaller than outer cell, anastomosing veins may be present inside of cells or arise from cells. Femora incrassate, ventral surface with larger setigerous tubercles, fore and mid tibiae each with fossula spongiosa occupying only apex of tibia. Lateral posterior angle of each connexival segment produced. Pygophore with posterior margin nearly straight, median process short; paramere short and clavate; phallus thick, dorsal phallothecal sclerite small and nearly separated from main part of phallosoma.

**Distribution:** Papua New Guinea.


**Key to the species in the genus *Tympanistocoris* Miller**
1. Posterior margin of seventh abdominal segment rounded (Figure 3A and Figure 11A); lateral margin of pronotum without distinct carina (Figure 2A) .............................. *T. humilis* Miller–. Posterior margin of seventh abdominal segment emarginated (Figure 8B and Figure 13A); lateral margin of pronotum with a distinct carina (Figure 6A) ........................... *T. usingeri* sp. nov.***Tympanistocoris humilis* Miller, 1958** (Figure 1, Figure 2, Figure 3 and Figure 4, Figure 10 and Figure 11)

*Tympanistocoris humilis* Miller, 1958: 227; Maldonado Capriles, 1990: 455.

**Redescription:***Coloration*. Yellowish brown suffused with darker irregular markings (Figure 1, Figure 10 and Figure 11). Most part of head beneath, median part of transverse sulcus on dorsum of head, apical half of first antennal segment, sub-basal portion of first visible labial segment, upper portion of basal and apical part of second visible labial segment, third visible labial segment, most anterior pronotal lobe, spot on each side of scutellum, most meso- and metasterna, most meso- and metapleura, most outer cell on membrane of hemelytron, annulations on fore and mid legs, and markings on lateral sides of abdomen dark brown; eyes black with irregular brown markings; ocelli yellowish; anterior portion of posterior pronotal lobe darker than posterior portion; spots on connexival segments black; veins pale and markings on hemelytron pale yellow.

*Structure*. Body covered with short, depressed pubescence and sparsely scattered with a few small setigerous tubercles except eyes, ocelli, head dorsally, anterior pronotal lobe dorsally, and membrane of hemelytron; parts beside inner margins of eyes, lateral sides and posterior portion of postocular portion, both ends of collar, lateral side of pronotum, lower surface of mid and hind femora with larger setigerous tubercles (Figure 2A,B,F,G). Head about 1.5 times as long as width across eyes, anteocular portion about 1.8 times as long as postocular portion (Figure 2B); tuberculate processes on dorsum of head nearly as long as half of first antennal segment (Figure 2C); interocular space nearly two times as long as interocellar space, interocellar space slightly shorter than distance between ipsilateral eye and ocellus (Figure 2B). First antennal segment (Figure 2C) slightly shorter than anteocular portion, remaining segments missing. First visible labial segment extending to middle of eye in lateral view (Figure 2A). Collar process with some large setigerous tubercles (Figure 2B). Pronotum about 1.4 times as long as wide; anterior pronotal lobe about 0.75 times as wide as posterior lobe, with distinct sculptures, four tuberculate processes longer than other dorsal processes on head and posterior pronotal lobe; posterior lobe of pronotum dorsally with four longitudinal carinae, two lateral discal carinae ending before anterior half of posterior pronotal lobe and two median carinae reaching dorsal processes before posterior margin; humeral angles of pronotum tuberculately produced and slightly pointed forwards, nearly as long as dorsal processes of head; posterior margin nearly straight (Figure 2B). Lateral margin of scutellum dorsally with an indistinct tubercule, apical part of scutellum destroyed by insect pin. Hemelytra slightly extending beyond apex of abdomen. Fore femur about nine times as long as its maximum diameter, slightly longer than fore tibia (Figure 2F). Abdomen nearly 1.3 times as long as width, connexival segments dilated, posterior angles of each connexival segment produced posteriad (Figure 9 and Figure 10). Pygophore nearly ovate in ventral view, posterior margin slightly concave medially (Figure 3D), median pygophore process short with apex rounded (Figure 3C). Paramere clavate, median part curved, inner side of subapical portion with a broad process (Figure 3E–H). Basal plate of phallus somewhat thickened, nearly straight (Figure 3I and Figure 4B); basal plate bridge short and thin, situated before half of basal plate (Figure 4B); pedicel thick and long (Figure 3I). Dorsal phallothecal sclerite small, slightly asymmetrical (Figure 4C,D), nearly totally separated from body of phallosoma (Figure 3I). Struts thin, nearly reaching apex of dorsal phallothecal sclerite (Figure 4C,D). Endosoma apically with a sclerotized cover, which, like the dorsal phallotheca of other reduviid species, middle of endosoma with two symmetrical strongly sclerotized bent structures (Figure 4E–G).

**Measurements** [in mm, male (n = 1)]: Body length, 11.3. Maximum width of abdomen (excluding posterior connexival segment processes), 4.2. Head length, 2.63. Length of anteocular portion of head, 1.34; length of postocular portion of head, 0.71. Interocular distance, 1.2. Interocellar distance, 0.52. Length of antennal segment, I = 1.32, II–IV: missing. Length of visible labial segments I: II: III = 1.07: 1.13: 0.45. Length of anterior lobe of pronotum, 1.02; length of posterior lobe of pronotum, 1.67. Length of hemelytron, 7.3.

**Type material:** Holotype male, “Type” [round, white with red circle, pr]; “Humboldt Bay, Malay Archipelago, W. Doherty; 1903-31” [rectangular, white, pr]; “New genus nr. *Caridomma*; det. RL Usinger’49” [rectangular, white, New genus nr. *Caridomma*: hw, det. RL Usinger’49: pr]; “*Tympanistocoris humilis* gen. n. sp. n. (holotype); N. C. E. Miller det. 1958” [rectangular, white, *Tympanistocoris humilis* gen. n. sp. n. (holotype): hw, N. C. E. Miller det. 1958: pr]; “BMNH_ENT, UCR_ENT 48516” [rectangular, white, pr]; “BMNH(E) 1517972” [rectangular, white, pr]. The specimen is not in good condition, one mid leg and first antennal segment are glued on the card together, antennae (except one basal segment), one mid leg, two hind legs, and part of hind wings are partly missing (Figure 10 and Figure 11).

**Female:** Unknown.

**Distribution:** Papua New Guinea, Humboldt Bay, known only from type locality for now.

**Remarks:** The gender of the holotype is misprinted as female in the original description of Miller [1]. The holotype is a male, and Miller illustrated the paramere of this specimen.

**Figure 2 insects-14-00165-f002:**
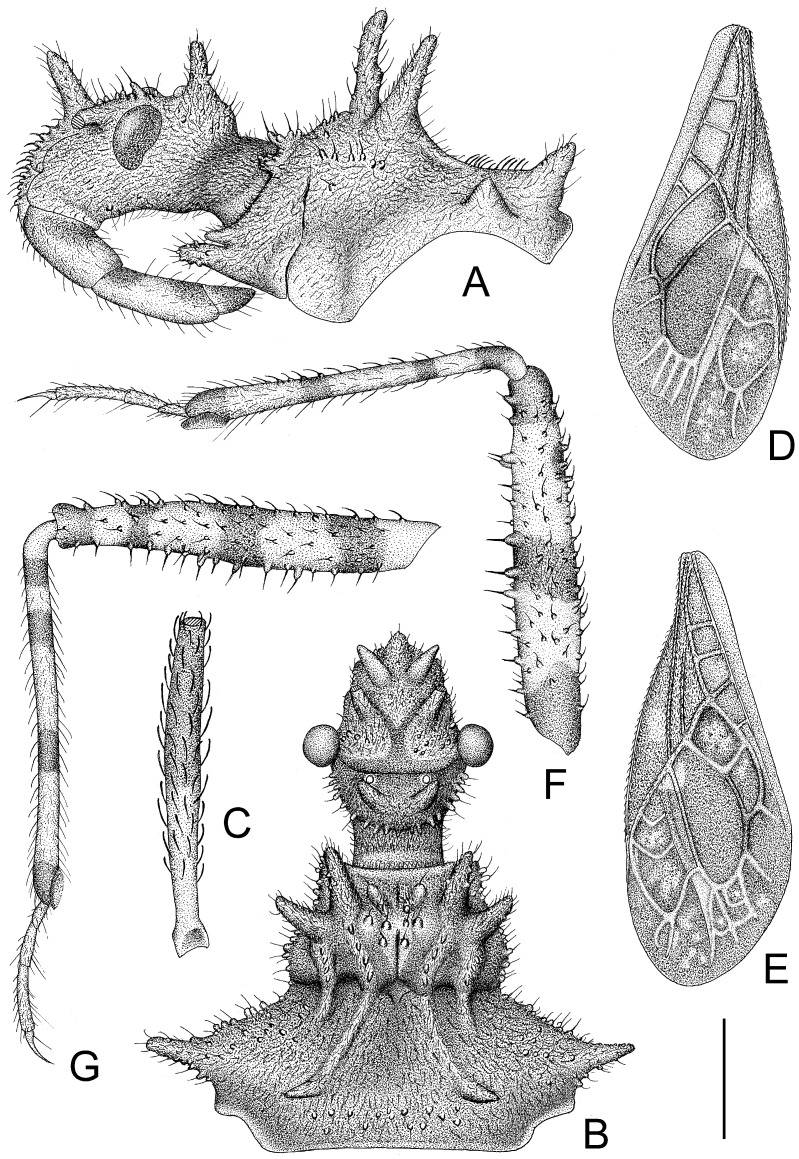
*Tympanistocoris humilis* Miller, male. (**A**,**B**), head and pronotum, antennae removed; (**C**), first antenna segment; (**D**), right hemelytron; (**E**), left hemelytron; (**F**), fore leg, coxa, and trochanter removed; (**G**), mid leg, coxa, and trochanter removed; (**A**,**F**,**G**), lateral view; (**B**–**E**), dorsal view. Scale bar for (**A**,**B**) = 0.9 mm, for (**C**) = 0.45 mm; for (**D**,**E**) = 2 mm; for (**F**,**G**) = 1 mm.

**Figure 3 insects-14-00165-f003:**
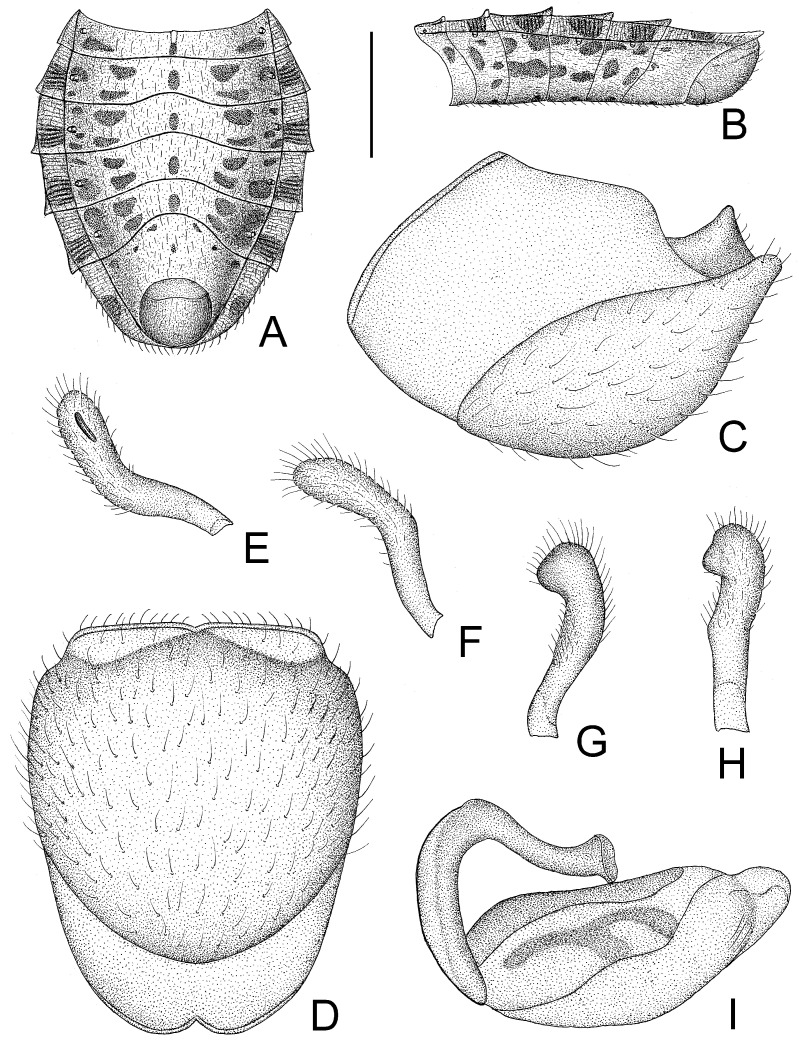
*Tympanistocoris humilis* Miller, male. (**A**,**B**), Abdomen; (**C**,**D**), pygophore; (**E**–**H**), paramere; I, phallus; (**A**,**D**), ventral view; (**B**,**C**,**I**), lateral view; (**E**–**H**), different views of same paramere. Scale bar for (**A**,**B**) = 2 mm, for (**C**–**I**) = 0.5 mm.

**Figure 4 insects-14-00165-f004:**
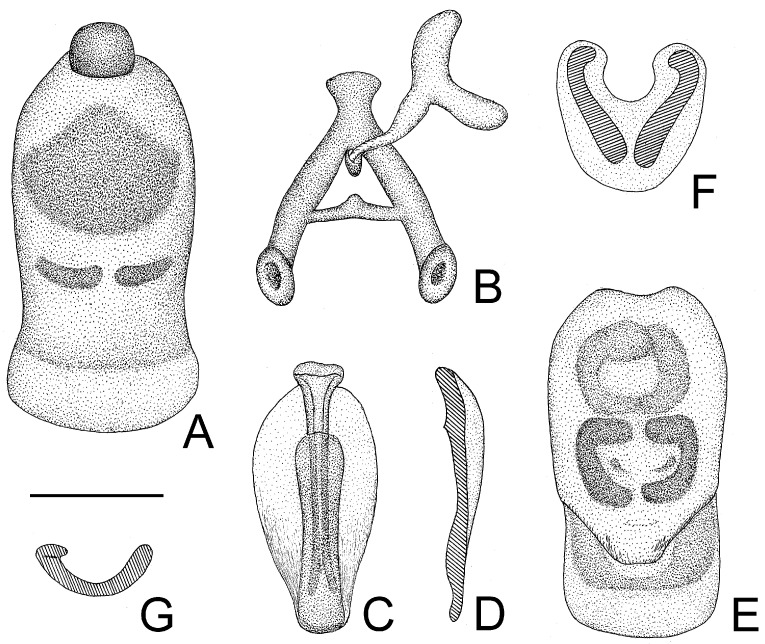
*Tympanistocoris humilis* Miller, male. (**A**), Phallus; (**B**), phallobase with bulbus ejaculatorius; (**C**,**D**), dorsal phallothecal sclerite and struts; (**E**), phallosoma, dorsal phallotheca, and struts removed; (**F**,**G**), processes of endosoma; (**A**), ventral view; (**B**,**C**,**E**,**F**), dorsal view; (**D**,**G**), lateral view. Scale bar = 0.5 mm.

***Tympanistocoris usingeri* sp. nov.** (Figure 5, Figure 6, Figure 7, Figure 8, Figure 9, Figure 12 and Figure 13)

**Diagnosis:** The new species can be easily distinguished from its congener, *T. humilis*, by the larger and darker body (vs. smaller and paler in *T. humilis*), the lateral sides of the pronotum with a distinct carina (Figure 6A) (vs. without distinct carina on the lateral sides of the pronotum in *T. humilis*, as shown in Figure 2A), and the posterior margin of the seventh abdominal segment emarginated (Figure 8B and Figure 13A) (vs. the posterior margin of seventh abdominal segment rounded in *T. humilis*, as shown in Figure 3A and Figure 11A).

**Description:***Coloration*. Brown suffused with darker and paler irregular markings (Figure 5, Figure 12 and Figure 13). Most part of head beneath, transverse sulcus on dorsum of head, most portion of first visible labial segment and third visible labial segment, most postocular portion, most anterior pronotal lobe, thoracic pleura and sterna, scutellum (except apical spine), most hemelytron, most femora, most abdominal beneath blackish; apical third of first antennal segment, most second antennal segment and apical two segments, dorsal part of base of second antennal segment, annulations of tibiae, and apices of third tarsomeres dark brown; most veins on membrane of hemelytron, portions of femora whitish yellow, most anteocular portion, apical part of first visible labial segment and most second visible labial segment, tibiae (except darker annulations), and markings ventrally on abdomen yellow; eyes reddish dark brown, ocelli reddish brown.

**Figure 5 insects-14-00165-f005:**
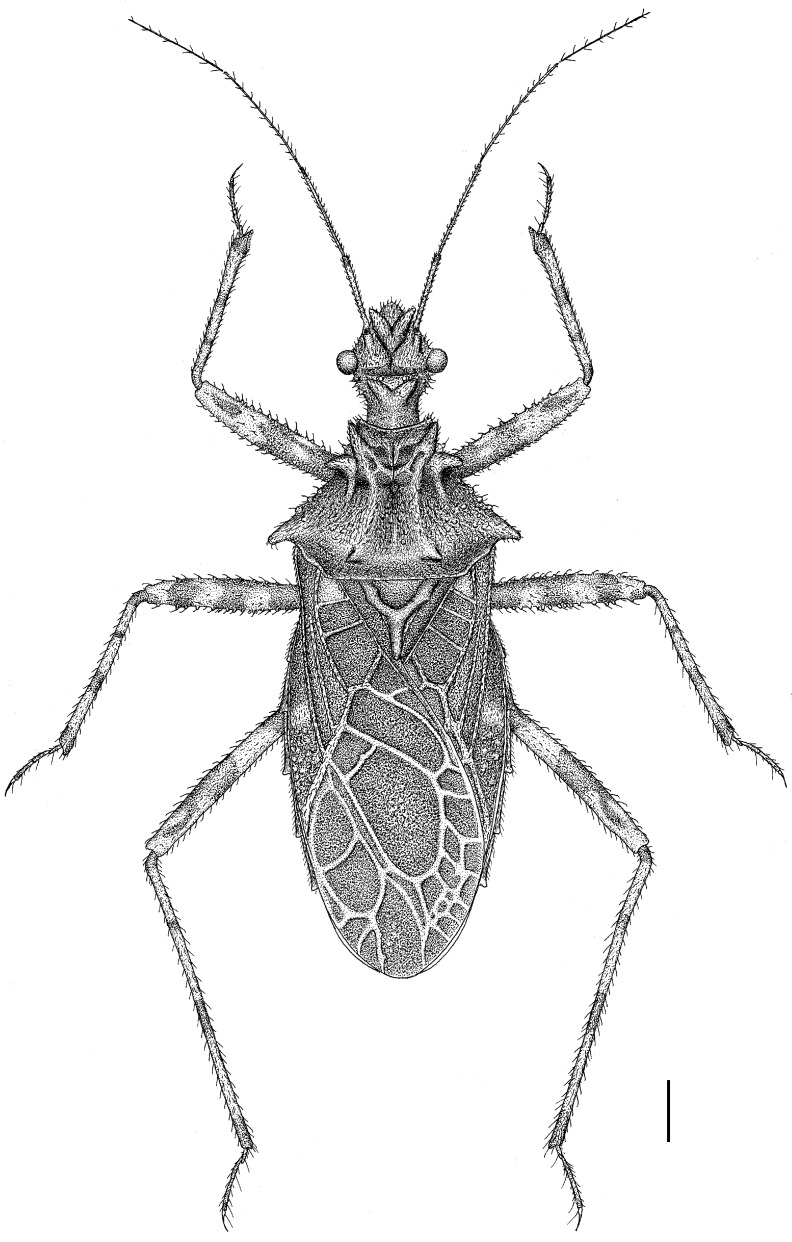
*Tympanistocoris usingeri*, sp. nov., male. Holotype, habitus. Scale bar = 1 mm.

**Figure 6 insects-14-00165-f006:**
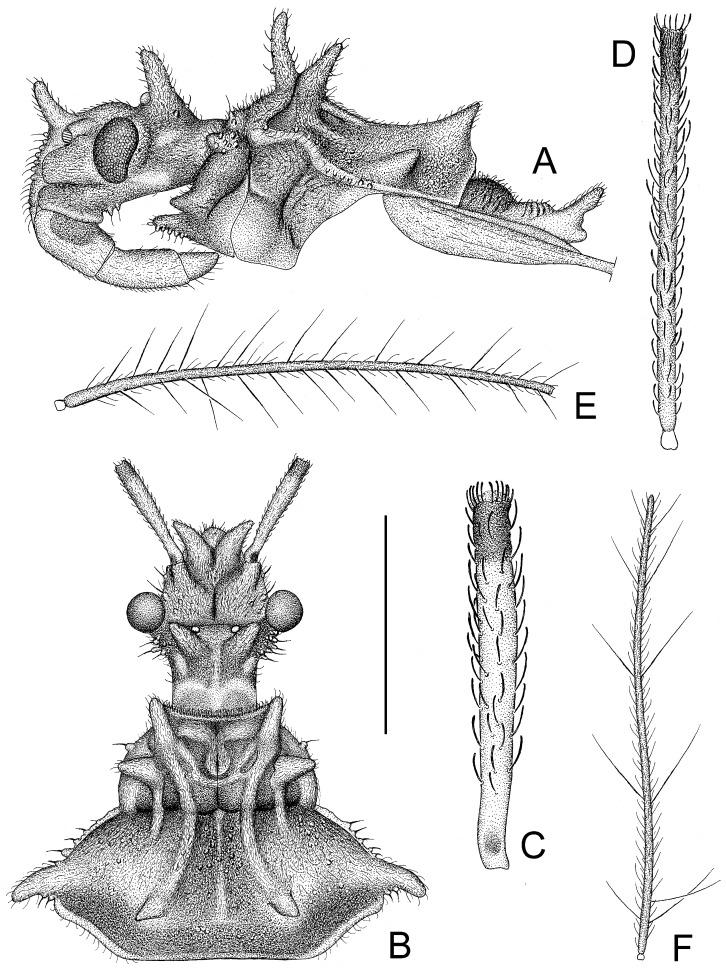
*Tympanistocoris usingeri*, sp. nov., male. (**A**), Head, pronotum, and scutellum, antennae removed; (**B**), head and pronotum, apical three antennal segments removed; (**C**), first antennal segment; (**D**), second antennal segment; (**E**), third antennal segment; (**F**), fourth antennal segment; (**A**), lateral view; (**B**–**F**), dorsal view. Scale bar for (**A**,**B**) = 1.5 mm, for (**C**–**F**) = 1 mm.

**Figure 7 insects-14-00165-f007:**
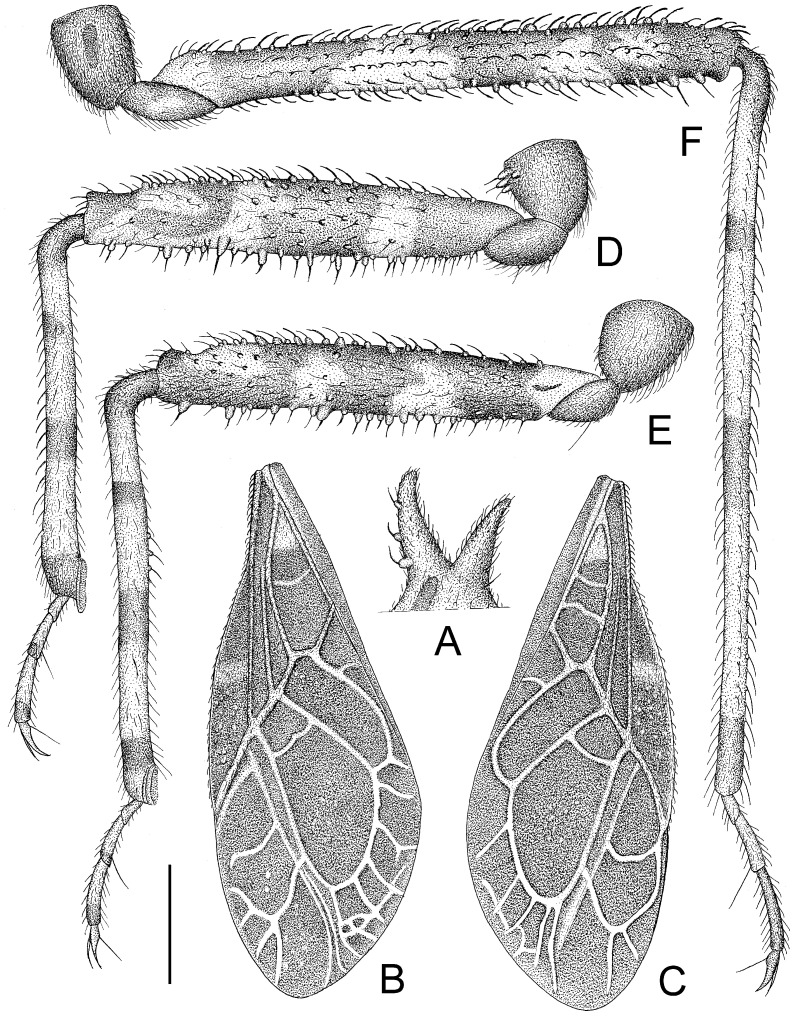
*Tympanistocoris usingeri*, sp. nov., male. (**A**), processes on anterior pronotal lobe; (**B**), left hemelytron; (**C**), right hemelytron; (**D**), fore leg; (**E**), mid leg; (**F**), hind leg; (**A**,**D**–**F**), lateral view; (**B**,**C**), dorsal view. Scale bar for (**A**,**D**–**F**) = 1 mm, for (**B**,**C**) = 2 mm.

*Structure*. Body covered with short, depressed pubescence and sparsely scattered with small setigerous tubercles except eyes, ocelli, sculpture on dorsum of head, sculpture on anterior pronotal lobe, and hemelytron; lateral sides of head, both lateral sides of collar, lateral sides of pronotum, anterior prosternal processes, trochanters, ventral surface of femora with larger setigerous tubercles (Figure 6A,B and Figure 7D–F). Head about 1.3 times as long as width across eyes, anteocular portion of head about 1.5 times as long as postocular portion (Figure 6B); tuberculate processes on dorsum of head shorter than half of first antennal segment; interocular space nearly 2.9 times as long as interocellar space, interocellar space slightly shorter than distance between ipsilateral eye and ocellus (Figure 6A,B).

**Figure 8 insects-14-00165-f008:**
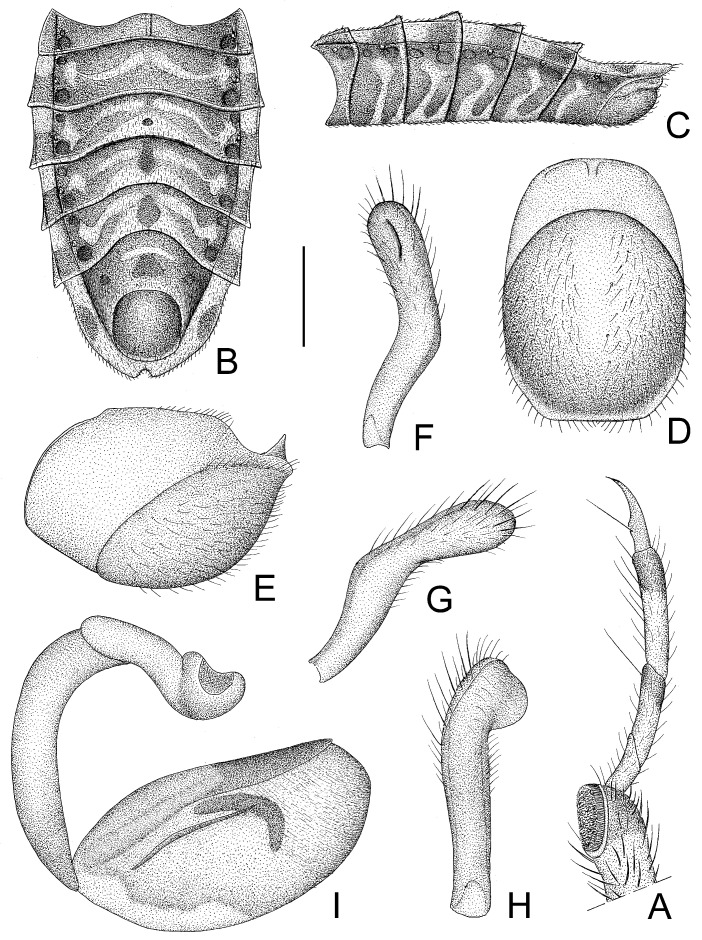
*Tympanistocoris usingeri*, sp. nov., male. (**A**), Apical part of fore leg; (**B**,**C**), abdomen; (**D**,**E**), pygophore; (**F**–**H**), paramere; (**I**), phallus; (**A**,**C**,**E**,**I**), lateral view; (**B**,**D**), ventral view; (**F**–**H**), different views of paramere. Scale bar for (**A**,**F**–**H**) = 0.5 mm, for (**B**,**C**) = 2 mm; for (**D**,**E**) = 1 mm.

First antennal segment (Figure 6C) distinctly longer than anteocular portion of head, second antennal segment (Figure 6D) about 1.1 times of first segment, third antennal segment (Figure 6E) nearly 1.5 times of first segment, fourth antennal segment (Figure 6F) slightly longer than second segment and about 1.2 times as long as first segment. First visible labial segment extending to middle of eye in lateral view (Figure 6B). Collar process with some large setigerous tubercles (Figure 6B). Pronotum about 1.44 times as long as wide; anterior pronotal lobe about 0.82 times as wide as posterior lobe, with distinct sculptures, anterior pair of tuberculate processes longer than other dorsal processes on head and posterior pronotal lobe (Figure 6B,C); posterior lobe of pronotum dorsally with six longitudinal carinae, lateral two situated at lateral margins of pronotum and ending at posterior margin of pronotum, two lateral discal carinae ending at 1/3 of posterior pronotal lobe, and two median carinae reaching dorsal processes before posterior margin; humeral angles of pronotum tuberculately produced and apex slightly pointed backwards, nearly as long as dorsal processes of head; dorsal processes on disc of pronotum short; posterior margin of pronotum nearly straight (Figure 6B). Basal 1/3 of lateral margin of scutellum dorsally with a tuberculate process, apical spine nearly as long as humeral angle process (Figure 6A). Hemelytra reaching apex of abdomen. Fore femur about six times as long as its maximum diameter, slightly shorter than fore tibia (Figure 7D); mid femur nearly as long as mid tibia (Figure 7E); hind femur about 0.8 times as long as hind tibia (Figure 7F). Abdomen nearly 1.5 times as long as width, connexival segments slightly dilated, posterior angles of each connexival segment produced upwards (Figure 8B,C); middle of posterior margin of seventh abdominal segment concave (Figure 8B). Pygophore ovate in ventral view (Figure 8D), posterior margin nearly straight, median pygophore process short and apex sharp in lateral view (Figure 8D,E). Paramere clavate with median part curved, apex rounded, inner side of subapical portion with a broad process (Figure 8F–H). Basal plate of phallus somewhat thickened, and short, middle portion nearly straight (Figure 8I and Figure 9A); basal plate bridge thin (Figure 9A); pedicel thick and long (Figure 8I). Dorsal phallothecal sclerite larger, nearly reaching apex of endosoma in resting condition (Figure 9B,D). Struts reaching apex of dorsal phallothecal sclerite (Figure 9D). Middle of endosoma with two symmetrical, strongly sclerotized bent structures (Figure 9B,E,F).

**Type material:** Holotype male, “Holo-type” [round, white with red circle, pr]; “♂” [rectangular, white, pr]; “On fallen log of *Spondias pinnata* air strip S.H. Dist B. Gray 26-xi-1969” [rectangular, white, hw]; “C. I. E. COLL. A. 5212” [rectangular, white, C. I. E. COLL. A.: pr, 5212: hw]; “791” [rectangular, white, hw]; “*Tympanistocoris* sp. M.S.K. Ghauri det. 1972” [rectangular, white, *Tympanistocoris* sp. hw, M.S.K. Ghauri det. 1972 pr]; “*Tympanistocoris usingeri* sp. n. Det. W. Cai, 2015” [rectangular, white, *Tympanistocoris usingeri* sp. n.: hw, Det. W. Cai, 2015: pr]; “BMNH(E) 1517973” [rectangular, white, pr]. The specimen is in relatively good condition with only left mid tarsus missing (Figure 10 and Figure 11).

**Female:** Unknown.

**Distribution:** Papua New Guinea, Orokana, known only from type locality for now.

**Etymology:** This species is named in honor of Dr. Robert L. Usinger, the late great heteropterist, who first recognized this special genus as new in 1949.

**Figure 9 insects-14-00165-f009:**
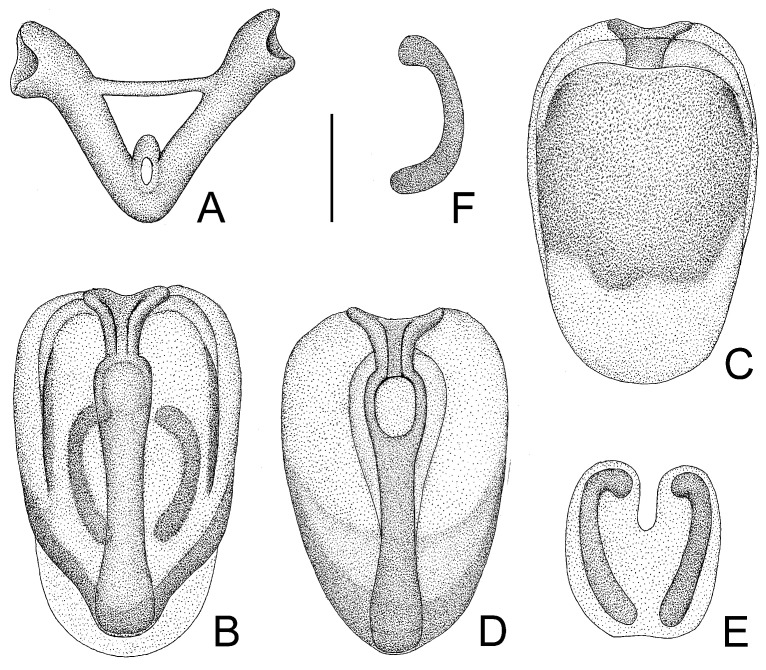
*Tympanistocoris usingeri*, sp. nov., male. (**A**), phallobase; (**B**,**C**), phallosoma; (**D**), dorsal phallothecal sclerite and struts; (**E**,**F**), processes of endosoma; (**A**,**B**,**D**,**E**), dorsal view; (**B**), ventral view, (**F**), lateral view. Scale bar = 0.5 mm.

**Figure 10 insects-14-00165-f010:**
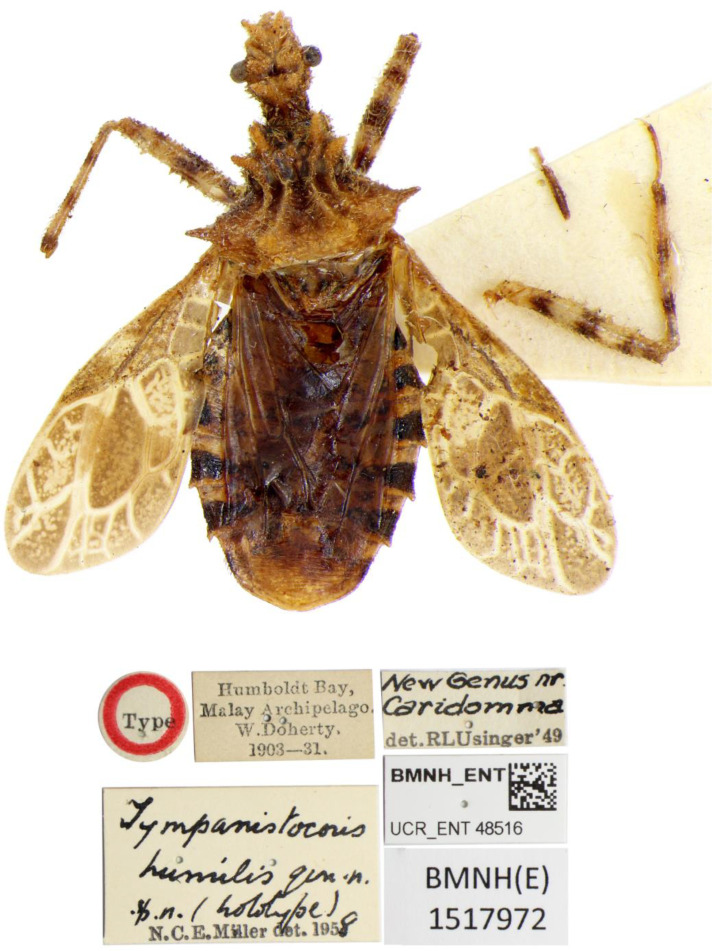
*Tympanistocoris humilis* Miller, male. Holotype, habitus in dorsal view, and labels.

**Figure 11 insects-14-00165-f011:**
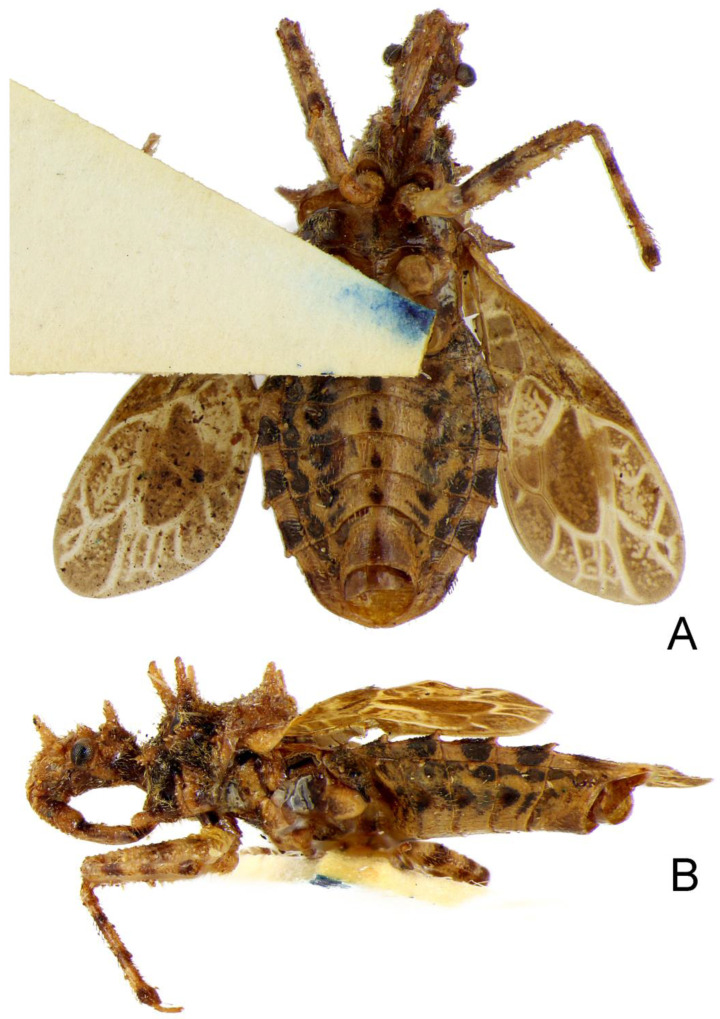
*Tympanistocoris humilis* Miller, male. Holotype, habitus, pygophore removed. (**A**), ventral view, (**B**), lateral view.

**Figure 12 insects-14-00165-f012:**
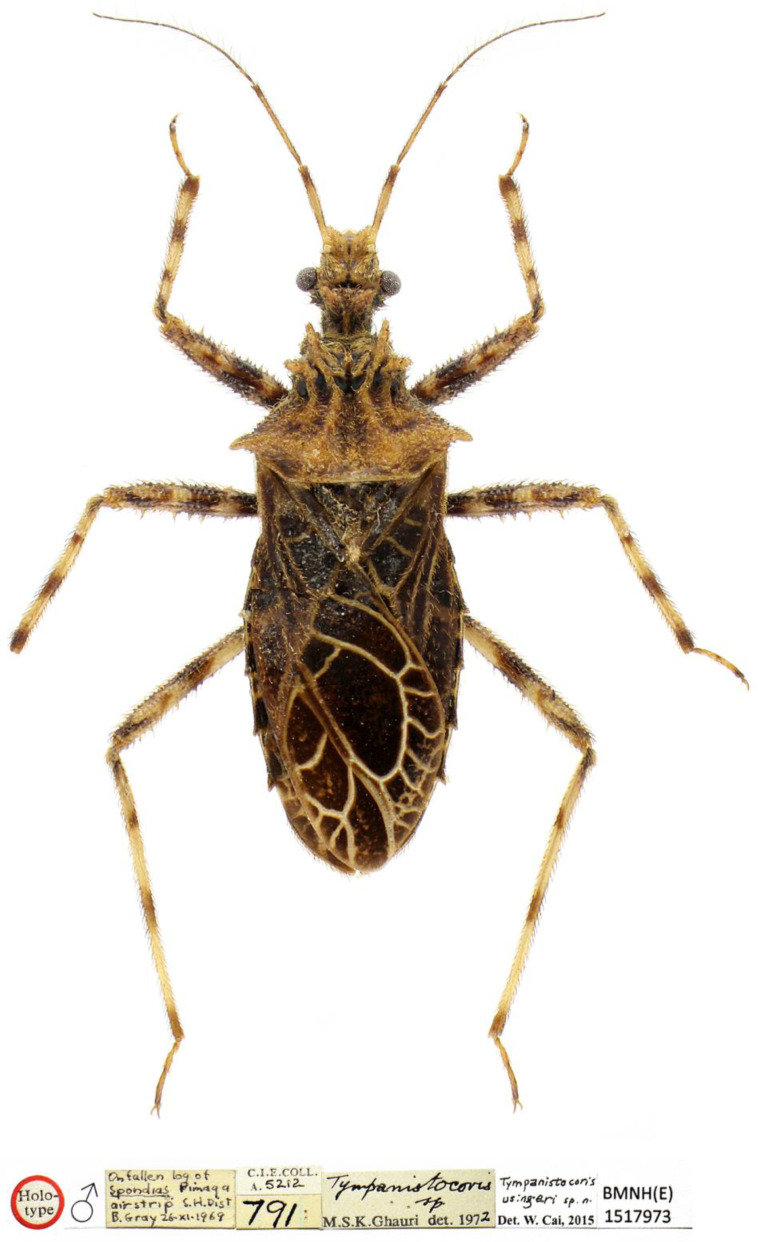
*Tympanistocoris usingeri*, sp. nov., male. Holotype, habitus in dorsal view, and labels.

**Figure 13 insects-14-00165-f013:**
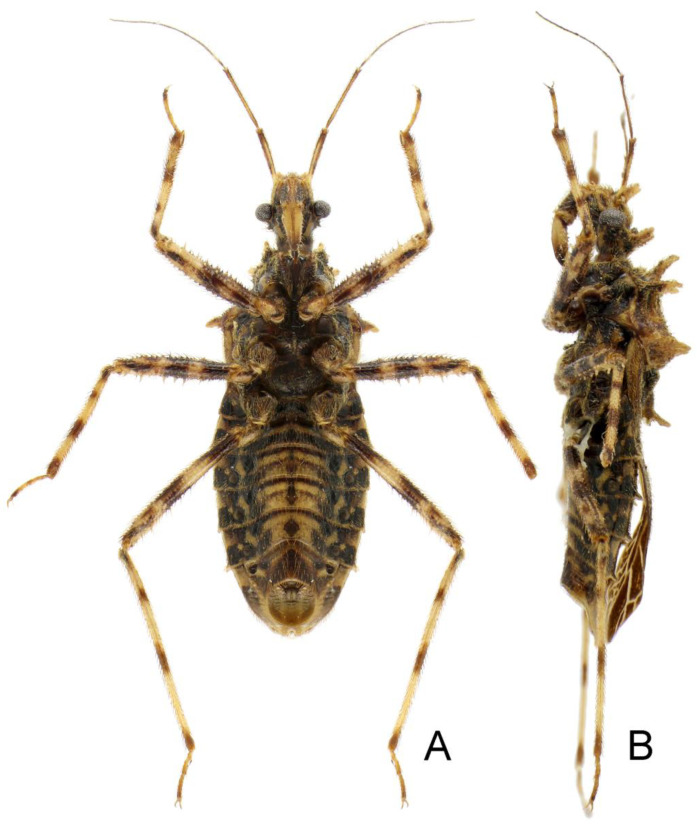
*Tympanistocoris usingeri*, sp. nov., male. Holotype, habitus, pygophore removed. (**A**), ventral view, (**B**), lateral view.

## 4. Discussion

### 4.1. Systematic Position of Tympanistocoris

This genus was firstly recognized as a new genus by the late heteropterist, Robert L. Usinger, in 1949 as one label of the holotype of *T. humilis* Miller, 1958 indicated that “new genus nr. *Caridomma*; det. R L Usinger’49”. As noticed by Usinger, the general body plan, the pronotal processes, and the coloration of this genus are similar to those of the cetherine genus, *Caridomma* Bergroth, from Africa to some extent. When establishing the genus, Miller noted that “The affinity of this new genus is doubtful. Its habitus somewhat suggests relationship with the subfamily Cetherinae, but, on account of the tuberculate head and thorax, the non-pedunculate eyes and the different kind of hemelytral venation, it cannot be correctly assigned to that subfamily. For these reasons I place it provisionally in the subfamily Reduviinae.” [3]. After the examination of the cetherine specimens, we found that the genus, *Tympanistocoris*, can be easily separated from *Caridomma*, as well as other cetherine genera by the long head and the transverse sulcus running behind the eyes (vs. the short head and the transverse sulcus running between the eyes in Cetherinae). Among the reduviine genera, *Tympanistocoris* closely resembles to the Australian genus, *Noualhierana* Miller, in the structures of head and pronotal, as mentioned above. The polyphyly of Reduviinae has been discussed by Weirauch [5] and Huang and Weirauch [6], which has greatly helped us to understand the diversity and evolution of Reduviinae. However, both genera, *Tympanistocoris* Miller and *Noualhierana* Miller, were not included in their phylogenetic analysis. The exact taxonomic position of *Tympanistocoris* in the Reduviinae needs further study based on different sources of data and the assistance of molecular analyses.

### 4.2. Anastomosing Veins on Hemelytra

In Reduviidae, the anastomosing veins have been reported in the phimophorine genus, *Phimophorus* Bergroth [7,8,9], and the triatomine genus, *Belminus* Stål [10]. The veinlets may present outside and inside of the two large cells on the membrane of the hemelytron, and Sandoval et al. recognized those veins as secondary veins [10]. The different individuals of *Phimophorus spissicornis* Bergroth may have different venations, as shown by Handlirsch [7], Usinger and Wygodzinsky [8], and Chaverra-Rodríguze et al. [9]. In the genus, *Tympanistocoris*, we find both species also have the anastomosing veins on the membrane of the hemelytron, and the venations of the left and right hemelytra of same species are asymmetric (Figure 2D,E, Figure 7B,C and Figure 10), especially those on the outside of the two large cells. Except for Reduviidae, the anastomosing veins have also been observed in other true bugs, such as species in *Megymenum* Guérin-Méneville (Dinidoridae), *Cryptorhamphus* Stål (Cryptorhamphidae), and *Spartocera* Laporte (Coreidae), as well as many genera in Aradidae [11,12,13].

Though the anastomosing veins are observed in both species in *Tympanistocoris*, this character has been rarely included in the generic diagnosis in Reduviidae because this structure may only be present in some species of the same genus (e.g., *Belminus*). Moreover, due to the asymmetry of the anastomosing veins on the left and right hemelytra, the presence, instead of the pattern, of these anastomosing veins could probably be the diagnostic character to distinguish different species. The similar color patterns of the hemelytron can be found in some species of Reduviidae in the subfamilies, Cetherinae (e.g., *Cethera* spp.) and Reduviinae (e.g., *Acanthaspis* spp.). It is easy to distinguish the color markings from the veins, but this could give us a hint about the putative function of the anastomosing veins. Generally, the heteropteran species with the anastomosing veins on the hemelytra mentioned above usually have the dull coloration (dark yellow, brown, black, etc.) and a relatively broad membrane. Moreover, these species mostly live in dim and covert environments; for example, one of the labels of the holotype of *T. usingeri*, sp. nov. says “On fallen log of *Spondias pinnata*”. As such, the complex and reticular veins or stripes could increase the camouflage ability and help the bugs better hide in their microhabitats, such as leaf litter or bark crevices.

## Data Availability

Data are contained within the article.

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
