# Peer review of "A Taxonomic Revision of the Assassin Bug Genus, Tympanistocoris (Hemiptera: Reduviidae: Reduviinae)"

_insects, 2023, doi:10.3390/insects14020165_

Round 1
Reviewer 1 Report
This paper taxonomically describes a new species of the assassin bug, Tympanistocoris usingeri, with the redescription of T. humilis. This is a taxonomic paper of high quality, and the figures of the bugs are elaborated and splendid. I think that there are no large problems in this paper.
Minor comments are,
Line 18, 110, and 193. “emarginted” should be “emarginated”.
Line 28. Italicize “Pheletocoris”.
Line 191. “by larger and darker” may be “by larger and darker body”.
Author Response
Dear reviewer:
Thank you so much for your comments!
Please see our response in the attachment PDF.
Best Regards

Reviewer 2 Report
1. Line 295 "Tympanistocoris is closely related to the Australian genus Noualhierana..." Authors should avoid invoking 'closely related' without a cladistic analysis. Instead, consider using 'closely resembles'
2. Line 296 "The polyphyly of Reduviinae have been discussed..." Typo. Change to "has" been
3. Line 297 "... and greatly helped us to understand the diversity and evolution of Reduviinae." Grammatically unsound. Consider changing to "The polyphyly of Reduviinae have been discussed by Weirauch and Huang & Weirauch, which has greatly helped us to understand the diversity and evolution of Reduviinae"
Author Response

(The authors gave the same response as above.)

Reviewer 3 Report
Dear authors and editor
The mauscript "A taxonomic revision of the assassin bug genus Tympanistocoris 2 (Hemiptera: Reduviidae: Reduviinae) " is a very valuable contribution on the genus Tympanistocoris giving solid evidence of a new species, the descriptions and figures are adequate, however there are some ambiguous terms that should be reconsidered, the figures I suggest to reorganize following the numbering.
Best regards

Author Response

(The authors gave the same response as above.)
